# Impacts of Low Temperature and Ensiling Period on the Bacterial Community of Oat Silage by SMRT

**DOI:** 10.3390/microorganisms9020274

**Published:** 2021-01-28

**Authors:** Xiaomei Li, Fei Chen, Xuekai Wang, Lin Sun, Linna Guo, Yi Xiong, Yuan Wang, Hongzhang Zhou, Shangang Jia, Fuyu Yang, Kuikui Ni

**Affiliations:** 1College of Grassland Science and Technology, China Agricultural University, Beijing 100193, China; Lixiaomei1008@sina.com (X.L.); 17683267025@189.cn (F.C.); xkwang2016@163.com (X.W.); sunlin2013@126.com (L.S.); linnam921@163.com (L.G.); xiongleslie@126.com (Y.X.); wangyuany5@163.com (Y.W.); zhouhz1998@126.com (H.Z.); shangang.jia@cau.edu.cn (S.J.); 2Inner Mongolia Academy of Agricultural & Animal Husbandry Sciences, Hohhot 010031, China

**Keywords:** low temperature, oat, silage, bacterial community, SMRT

## Abstract

The objective of this study was to investigate how storage temperatures influence the bacterial community of oat silage during the ensiling process via PacBio single molecule, real-time sequencing technology (SMRT). Forage oat was ensiled at four different temperatures (5 °C, 10 °C, 15 °C, and 25 °C) and ensiling days (7, 14, 30, and 60 days). With the rise in storage temperature, the lactic acid content showed an increased trend. Acetic acid production was observed highest in silage fermented at 5 °C compared with other treatments, and *Enterococcus mundtii* was also the dominant bacterial species. *Lactiplantibacillus pentosus* and *Loigolactobacillus rennini* were exclusively detected in silages at 10 °C, 15 °C, and 25 °C, and dominated the fermentation after 60 days of ensiling at 10 °C and 25 °C, respectively. In addition, *L. pentosus*, *L. rennini*, and *E. mundtii* may be related to changes in the fermentation products due to the differences in ensiling temperature. In conclusion, results of this study improve our understanding of the complicated microbial composition underlying silage fermentation at low temperatures, which might contribute to target-based regulation methods for enhancing silage quality and developing new inoculants.

## 1. Introduction

Ensiling has been used for preserving the fresh forage for decades, especially in the regions where animals need to be wintered for a long period [1]. The ensiling process mainly relies on the epiphytic microorganisms especially lactic acid bacteria (LAB), which could determine the fermentation quality [2,3]. The microbiota ecosystem of optimal enisling is usually dominated by LAB with *Lactobacillus* species being the most frequently found [2]. In recent years, there has been a significant increase in the knowledge of new species and the metabolism of different species and strains, which has enhanced our knowledge about the presence and performance of microbiota in the process [4]. The improper ensiling includes a large number of undesirable microorganisms, which are detrimental for the nutrition value of silage [5]. Therefore, it is critical to classify the microbial diversity in silage.

Temperature is an important factor affecting silage quality, especially in hot or cold regions [6,7]. Generally, the temperature ranging from 20–30 °C is considered to be preferred for silage fermentation [7]. The microbial dynamics of microbial community of silage has been studied for the high temperature [8,9]. However, to our knowledge, only very limited studies were conducted on the effect of low temperature on the silage microbiota [7]. Harvesting forages in the cold region, such as the north of China, makes these forages take a longer time for achieving ideal fermentation quality. Previous studies speculated that the low fermentation quality in cold regions was due to the low temperature negatively influencing the microbial activity during the ensiling process [7]. Besides, some added silage inoculants can also be impaired, as the strains were often selected at the temperature associated with warm climate [6].

Ensiling has been an important way for preserving fresh oat for a long period in the world [10]. In the last decades, the planting area of oat has expanded toward to north regions in China, where the daily mean temperature ranges from 0 °C to 15 °C. The low temperature could lead to poor fermentation quality and higher nutrition losses. Although the effect of low temperature-adapted LAB strains on the oat silage quality have been investigated, they did not consider the bacterial community at the low temperature condition. Recently, next-generation sequencing (NGS) has been utilized to quantify and analyze silage microbiota [11,12]. Shorter sequences with relatively low taxonomical resolution have limited the classification of microorganisms in the community to the genus level [13]. The method of PacBio single molecule, real time sequencing technology (SMRT) could classify the bacteria into species level because of its relatively high taxonomic resolution [14,15], which could help us seek the certain species adapted to low temperature. Unfortunately, until now, relevant studies are seldom. 

Therefore, the objective of this study was to investigate the effect of temperature ranging between 5 °C and 25 °C on the bacterial community and fermentation quality during the process of oat ensiling by SMRT.

## 2. Materials and Methods 

### 2.1. Forage and Ensiling

Forage oat (*Avena sativa*, Mengyan No.3) was cultivated in the experimental field of Inner Mongolia Academy of Agricultural & Animal Husbandry Sciences, Wuchuan, China, which was sown in May 25, 2019 with a five-row planter with 0.20 m row spacing at a seeding rate of 150 kg/ ha. At the tillering stage, the field was fertilized with 150 kg/ha of urea (N > 46%) and no herbicides were applied during the whole growth period. On September 1st, 2019, the fresh forage oat was harvested at the boot stage with 4–5 cm stubble height and then directly chopped to the theoretical length of 2–3 cm using a crop chopper (ZS-2, Zhongsheng agricultural machinery company, Tangshan, China) without wilting. Then 500 g chopped oat material was mixed homogenously and packed manually into 35 cm × 50 cm polyethylene bags and vacuumed tightly. A total of 48 bags (4 temperatures × 4 ensiling days × 3 replicates) were prepared and were equally divided and kept at four different temperatures (5 °C, 10 °C, 15 °C, and 25 °C). Silages were opened and sampled at the ensiling time of 7, 14, 30, and 60 d, silages were opened to evaluate their fermentation end products and microbial community.

### 2.2. Fermentation Quality and Chemical Composition Analysis

The silage samples (20 g) were blended with 180 mL sterilized water and then stored at 4 °C for 24 h [11]. The filtrate used to measure fermentation quality of oat silage was performed with 0.22 µm filter paper [11]. The pH value was determined with a glass electrode pH meter (PHS-3C, INESA Scientific Instrument, Shanghai, China). The concentration of organic acid including lactic acid, acetic acid, propionic acid and butyric acid were determined by high-performance liquid chromatography (HPLC) (column: Shodex RS Pak KC-811; Showa Denko K.K., Kawasaki, Japan; detector: DAD, 210 nm, SPD-20A; Shimadzu Co., Ltd., Kyoto, Japan; eluent: 3 mmol L^−1^ HClO4, 10 mL min^−1^; temperature: 50 °C). 

About 200 g of each pre-ensiled material and silage sample were dried at 65 °C for 48 h by oven to test DM (dry matter) content, and then milled to pass through a 1.0 mm screen for determination of chemical composition. The contents of OM (organic matter) and EE (ether extract) were analyzed according to AOAC [16]. Total nitrogen (TN) content was determined by the Kjeldahl procedure (FOSS KjeltecTM 2300) and CP (crude protein) was calculated by multiplying TN with 6.25 [16]. NDF (neutral detergent fiber) and ADF (acid detergent fiber) contents were determined by the method of Van Soest et al. [17]. 

### 2.3. Microbial Population Analysis by Plate Culture 

Twenty grams of pre-ensiled materials and silage samples were immediately blended with 180 mL sterilized water and serially diluted from 10^−1^ to 10^−6^. Then, 20 µL of dilution were spread onto the corresponding agar separately. The microbial population were measured by plate culture method [2]. LAB were cultured onto de Man, Rogosa, Sharpe (MRS) agar medium and kept in anaerobic incubator at 30 °C for 48 h. Molds and yeasts were cultured onto a general incubator on Rose Bengal agar mediums at 28 °C for 48 h. Coliform bacteria were cultured onto Eosin-Methy Blue agar medium at 37 °C for 48 h. All mediums were obtained from Beijing Aoboxing Bio-tech Co., Ltd., Beijing, China. Colonies were counted as viable numbers of microorganisms in colony forming units (cfu)/g of fresh matter (FM). 

### 2.4. Microbial Community Analysis

#### 2.4.1. DNA Extraction and Bacterial Composition Analysis by SMRT Method

Ten grams of samples were mixed with 90 mL of sterile 0.85% NaCl solution and then vigorous shaken at 120 rpm, 4 °C for 2 h. The sample was filtered with two layers gauze. The filtered fluid was centrifuged at 10,000 rpm, 4 °C for 10 min. The deposit was suspended in 1 mL of sterile 0.85% NaCl solution, and then was centrifuged at 12,000 rpm, 4 °C for 10 min. The total DNA was extracted from resulting cell pellets. The quality and concentration of the extracted DNA was checked via 1% agarose gel electrophoresis and spectrophotometry (optical density at 260/280 nm ratio). All the DNA samples were purified through the DNA kit column (DP214-02, Tiangen, Beijing, China) and kept at −20 °C until further analysis.

The DNA was amplified by the bacterial 16S rDNA primers targeting the V3-V4 regions of 338F (5′-ACTCCTACGGGAGGCAGCA-3′) and 806R (5′-GGACTACHVGGGTWTCTAAT-3′). PCR condition was set as follows: initial denaturation at 98 °C for 2 min, denaturation at 98 °C for 30 s, 30 cycles of annealing at 50 °C for 30 s, elongation at 72 °C for 60 s, and extension at 72 °C for 5 min. Each treatment was conducted in triplicate, and the mixture was purified. Sequencing and analyzing were performed according to Guo et al. [18].

#### 2.4.2. SMRT Sequencing of Bacterial Diversity

The full-length 16S ribosomal RNA (rRNA) gene was amplified by PCR for SMRT sequencing using the forward primers 27F (5′-AGRGTTTGATYNTGGCTCAG-3′) and the reverse 1492R (5′-TASGGHTACCTTGTTASGACTT-3′). The PCR program was as follows: 95 °C for 5 min; 30 cycles of 95 °C for 30 s, 55 °C for 30 s, and 72 °C for 90 s, with a final extension of 72 °C for 7 min. Sequence pre-processing were performed on a PacBio Sequel platform (Pacific Biosciences, Menlo Park, CA, USA). Raw Barcode-CCS sequence data was obtained with lima v1.7.0 software. Then filtering raw Barcode-CCS to get valid sequence. Using the Quantitative Insights into Microbial Ecology (QIIME) package (version 4.2) to remove the low-quality sequence. The unique sequence set was classified into OTUs based on a 97% threshold identity using UCLUST [19]. Subsequently, representative sequence was compared using the Mothur3 software with the Silva database to gain classified information [20]. Functional genes of the bacterial communities were predicted using phylogenetic investigation of communities by reconstruction of unobserved states (PICRUSt) [21]. The final predicted metagenome was analyzed by Statistical Analysis of Taxonomic and Functional Profiles (STAMP) v.2.1.3 [22] Cluster analysis produced using R software (ver. 3.2.5) based on the OTUs. Prior to the redundancy analysis (RDA) was analyzed using detrended correspondence analyses that were conducted by the R software package4 (ver. 3.2.5) [23].

### 2.5. Statistical Analyses

Chemical composition, fermentation quality, and microbial counts data were analyzed using a two-way analysis of variance, with ensiling period and ensiling temperature as the main variables. The level of statistical significance was set to *p* < 0.05 using the SAS program version 9.1 (SAS Institute, Cary, NC, USA).

## 3. Results and Discussion

### 3.1. General Characteristics of Fresh Oat 

As shown in Table 1, the DM content of fresh oat was 23.70%, which was similar to the report by Jia et al. and Gomes et al. [10,24]. The contents of CP, ADF, and NDF were 13.25, 32.80 and 49.64% of DM, respectively. The CP in this study was higher, whereas the ADF and NDF contents were lower compared with the report by Wang et al. [25]. It was because several factors could influence the forage nutrition, including plant genotype, sowing density, harvest time and so on [26]. The population of LAB, yeast and coliform in fresh forage were 3.61, 4.37 and 4.10 log10 cfu/g FM^−1^, respectively. Interestingly, the number of the microbial population was nearly 100 times less compared with the data by Wang et al. [25]. That discrepancy was probably because the employed oat forage was planted at the lower temperature region.

### 3.2. Chemical Composition and Microbial Population by Plate Culture of Oat Silage

The chemical composition and microbial population of oat silage after 60 days of ensiling were shown in Table 2. Although all temperature treatments did not show significant difference on the contents of DM and NDF, significant impacts on the CP and ADF were observed. The observed less CP content in low temperature (5 °C and 10 °C) was similar to the report by Zhou et al. [7]. That could be ascribed to the insufficient silage fermentation in lower temperature, such as delayed acidification and lower rate of pH decline (Figure 1). However, we could not classify the reason behind why lower ADF of silages also appeared at 5 °C. Silages fermented at 25 °C had lower LAB count than other treatments, which could be the final lower pH values inhibiting the growth of LAB. Yeast is often considered as the primary initiators for causing aerobic spoilage. In this study, yeast and molds were not detected in all silages after 60 days of ensiling. Usually, the growth of yeast can be inhibited under anaerobic condition, especially after long time storage. Besides, the production of organic acid could further suppress its growth. Coliform bacteria have often been identified in alfalfa, corn and forage soybean silages [11,27], and its presence is usually associated with nutrition loss. Although a significant difference in coliform bacteria counts was found between different temperature treatments, their counts were detected at a low level, ranging around 10^2^–10^3^ cfu/g FM^−1^. Besides, at such low temperatures, the metabolic activity of coliform bacteria could be very low [28].

### 3.3. The Changes of pH and Organic Acid during Ensiling Process

The changes of pH and organic acid during ensiling process were shown in Figure 1 and Figure 2, respectively. As we known, the rate of pH decline is considered as an important indicator for reflecting the microbial activity and silage fermentation quality. In this regard, our results confirmed that the low temperature could restrict the silage fermentation. Because Figure 1 clearly showed that the rate of pH decline at 25 °C was higher than other treatments. Besides, at the temperature 5 °C, the silage acidification was greatly delayed, and the rate of decline was the slowest among all the treatments. Zhou et al. reported that sufficiently low pH values were eventually reached even if whole-plant corn silages were stored at 5 °C or 10 °C [7]. However, the final pH values at the same storage temperature were still around 5–6 after 60 days of ensiling in this study. This could be due to the good ensilage properties of whole-plant corn. In addition, a decline trend of pH was observed in silages at 5 °C and 10 °C during the whole ensiling time, which might indicate that prolonging storage time can contribute to the acidification of oat silage at low fermentation temperature. 

Lactic acid and acetic acid were the dominant fermentation products in all silages, but propionic acid and butyric acid were not detected. Silages stored at higher temperatures (15 °C and 25 °C) produced higher lactic acid than at 5 °C and 10 °C. It was observed that a great difference of pH and lactic acid content appeared between 5–10 °C and 15–25 °C, indicating that the temperature ranging from 10 °C to 15 °C could be the boundary for oat silage fermentation. Unexpectedly, relatively high acetic acid content was found at 5 °C, which was comparable to its lactic acid content. Lactic/acetic acid ratio close to 1:1 suggested hetero-fermentation, and this finding perhaps indicated that lower temperature leads to the fermentation pattern towards hetero-fermentation in oat silage. According to the bacterial community of oat silage (Figure 4), *Levilactobacillus brevis* was one of the bacterial indicators in 5 °C silage, and its hetero-fermentative pattern could account for the reason why higher acetic acid was detected at lower temperature. Adversely, the highest acetic acid content was found at the whole-plant corn silage with higher storage temperature [7]. This discrepancy could be attributed to the difference of microbial community in oat and whole-plant corn silages, especially LAB. For example, *Leuconostoc citreum*, *Latilactobacillus sakei* and *Latilactobacillus curvatus* prevailed in whole-plant corn silage at 5 °C and 10 °C, while all of them were not detected at the oat silage. 

### 3.4. Bacterial Community during Ensiling Process

As listed in Table 3, the coverage values of all samples were around 0.99, indicating the credible analysis of the microbial composition. The OTUs number per sample ranged from 30 to 60. Chao, as another richness index of bacterial community, showed a similar trend with OTUs. Shannon index varied between 0.5 and 2.1, and it could reflect the diversity index of microbial population. Both the factors of temperature (T) and ensiling day (D) had a significant impact on the Shannon diversity. Generally, with the advance of ensiling time, the establishment of aerobic and acidic environment will suppress the growth of most bacteria, leading to lower bacterial diversity. That is constant with our finding in oat silages at 5 °C, 15 °C, and 25 °C, in which a decrease trend of bacterial diversity was observed, while it showed an increased trend at 10 °C. 

The bacterial community of oat silage at the genus level was shown in Figure 3. Results of our research clearly demonstrated the different profiles of bacterial population diversity during ensiling process in relation to storage temperature. The main genera in 5 °C silages were *Enterococcus* and *Pantoea*, accounting for beyond 70% of total bacteria. At 10 °C, *Enterococcus* and *Hafnia-Obesumbacterium* were the dominant bacteria at the primary period, then replaced by *Lactobacillus* and *Enterococcus*. *Enterococcus*, a kind of cocci-shaped LAB, is usually present in silage fermentation. Several studies have shown *Enterococcus* played an important role in accelerating the silage fermentation, but *Enterococcus* could survive only in the early stages of fermentation due to its non-acid resistance characteristic [2,11]. Therefore, its high abundance means the low lactic acid content in silage, which is similar to our finding in silage at 5 °C. *Lactobacillus* play a critical role in reducing silage pH at the later stage. After 60 days of ensiling, *Lactobacillus* abundance became the highest among all the general at 10 °C, while its lactic acid content was much lower than at 15 °C and 25 °C. However, the lactic acid content at 10 °C presented constant increase trend during the whole ensiling process. That indicated if the time for oat ensiling at 10 °C was prolonged, the lactic acid content will continue to rise with the motivation of *Lactobacillus*. *Pantoea* was mainly detected in pre-ensiled fresh materials, such as forage soybean, guinea grass, and alfalfa, but *Hafnia-Obesumbacterium* was seldom found in fresh forage or silage. Until now, the roles of *Pantoea* and *Hafnia-Obesumbacterium* during ensiling process have not been extensively studied. 

*Lactobacillus*, *Enterococcus* and *Enterobacter* were the mainly abundant genus at 15 °C. *Lactobacillus* and *Enterobacter* were the dominant genus at the first 14 days of ensiling at 25 °C, then *Enterobacter* abundance decreased to a marginal level. The higher *Lactobacillus* abundance in 15 °C and 25 °C silages supported the higher increase rate of lactic acid. *Enterobacter* are non-spore forming, facultative anaerobe and could ferment lactic acid to acetic acid and other products, thus cause nutrition loss. Interestingly, silages at 15 °C had much higher *Enterobacter* abundance than at 10 °C, even though higher lactic acid content was found at 15 °C. Besides, silages at 25 °C, which possessed the highest lactic acid content, had very low *Enterobacter* abundance. The above results might suggest that sufficient lactic acid concentration or relatively low storage temperature could efficiently inhibit the growth of *Enterobacter* in oat silage.

The relative abundance of silage bacteria at the species level was shown in Figure 4. *E. mundtii* was the predominant species in silages at 5 °C, and also dominated in the silages at 10 and 15 °C. Bigwood et al. observed that *E. mundtii* could control the growth of *Listeria monocytogenes* at low temperatures (5 °C and 10 °C), indicating a potential application in chilled foods [29]. Klein reported that *E. mundtii* is mostly isolated from plant-related samples, thus, this species produces bacteriocins that are expected to be especially suitable for preservation of plant-related fermentation products [30]. These results showed that *E. mundtii* not only could adapt to the low temperature environment in oat silage, but also have the inhibitory ability against undesirable microorganism. In this regard, the role of *E. mundtii* in silage fermentation of low temperature deserves further study.

*L. pentosus* was exclusively identified in oat silage at 10 °C, 15 °C, and 25 °C, and became to be the most abundant species at 10 °C after 60 days of ensiling. Agarussi et al. have performed the trail of inoculating silage with *L. pentosus*, and the results showed that it had strong ability of producing lactic acid and reducing pH [31]. According, *L. pentosus* might be an interesting species for the development of LAB inoculants for cold climates. *L. rennini* quickly prevailed and could be identified as indicator species for oat silage at 25 °C. *L. rennini* is homo-fermentative bacteria and cannot grow at the temperature of <13 °C and >45 °C [32]. From our knowledge, this is the first report of *L. rennini* occurring in silages. Additionally, few studies have explored the function of *L. rennini*. In this study, the rich abundance of *L. rennini* in oat silage at 25 °C suggested that it could explain the high lactic acid content, but its effect at low temperature was weak. 

In line with the prevalence of *Enterobacteriaceae* is the finding that *Enterobacter* species were frequently detected in oats silages, such as *E. cancerogenus* and *E. cloacae*. More notably, *E. cancerogenus* accounted for 28.99% of bacterial population on the last day of ensiling at 15 °C, and for silage at 25 °C, its abundance nearly reached 40% at the first 14 days of ensiling. Pérez-Díaz et al. [33] has reported that *E. cancerogenus* and *E. cloacae* are able to convert lactic acid to propionic, and butyric acids, leading to an increase in pH in fermented food. Li et al. detected the presence of *E. cloacae* when the ensiling period of Guinea grass was prolonged [34] and suggested that the detected *Enterobacteria* could be associated with large amounts of 2,3-butanediol and ethanol produced. Although we did not detect the above phenomenon in our study, the potential hazards on silage quality need to be cautioned. Importantly, further research is needed to test the role of *E. cancerogenus* in silage fermentation. If the presence of Gram-negative bacteria *E. cancerogenus* significantly impairs fermentation quality of silage, there will be more reasons to eliminate and/or control its growth especially at the early stage of the silage fermentation.

### 3.5. Cluster Analysis of the Bacterial Community and Its Correlation with Fermentation Products

Cluster analysis was performed in order to acquire an overview of bacterial community at different storage temperature and ensiling time (Figure 5). Basically, the silage samples at 5 °C can be well separated from other treatment silages, indicating that little variation in bacterial composition occurred during ensiling process of oat at 5 °C. For silage at 25 °C, the samples ensiled for 7 and 14 days were clearly distinguished from at 30 and 60 days. It could be ascribed to the sudden increase of *L. rennini* abundance after 14 days of ensiling. Based on the RDA analysis (Figure 6), the obvious relationship between bacterial community at a species level and major fermentation products are visualized. *L. rennini* had a far position distance with *L. pentosus* and *Enterobacter* species, indicating that *L. rennini* might be a crucial indicator for affecting silage fermentation. The far distance among different bacterial species indicated that temperature had a significant impact on silage quality and microbial community.

## 4. Conclusions

The data from our study showed that the storage temperature influenced the bacterial species diversity and fermentation parameters of oat silage with notable variation along the temperature gradient. Also, it has been clearly indicated that a change in bacterial community during the ensiling process occurred in all treatment silages except at 5 °C. *E. mundtii* and *L. rennini* were the typically bacterial species in low and high temperature silages, respectively. Besides, their abundance may be associated with the changes of fermentation products of oat silage. This study provides new insights into the bacterial role in low temperature silage fermentation, which will advance understanding of fermentation of forage crops at a low temperature.

## Figures and Tables

**Figure 1 microorganisms-09-00274-f001:**
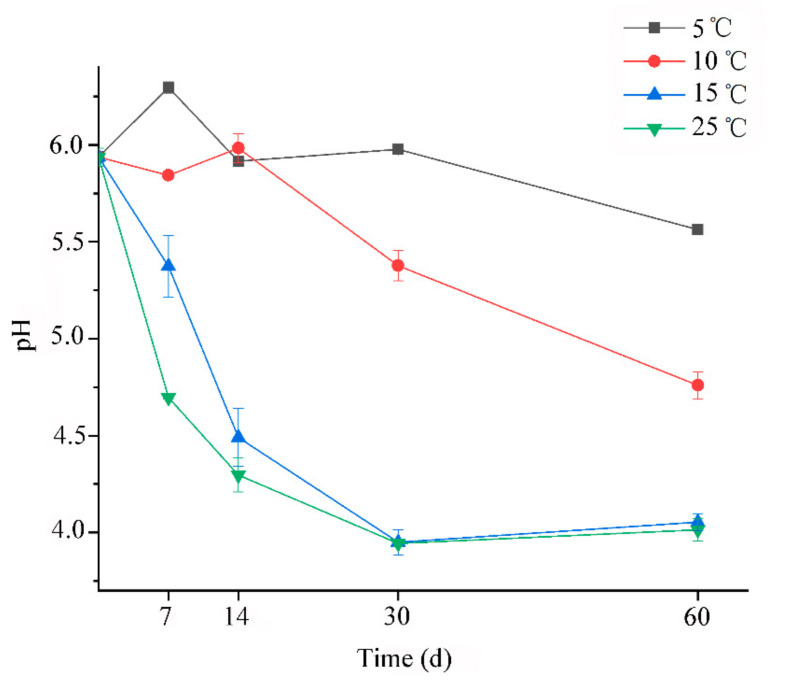
The change of pH for oat silages at different temperatures.

**Figure 2 microorganisms-09-00274-f002:**
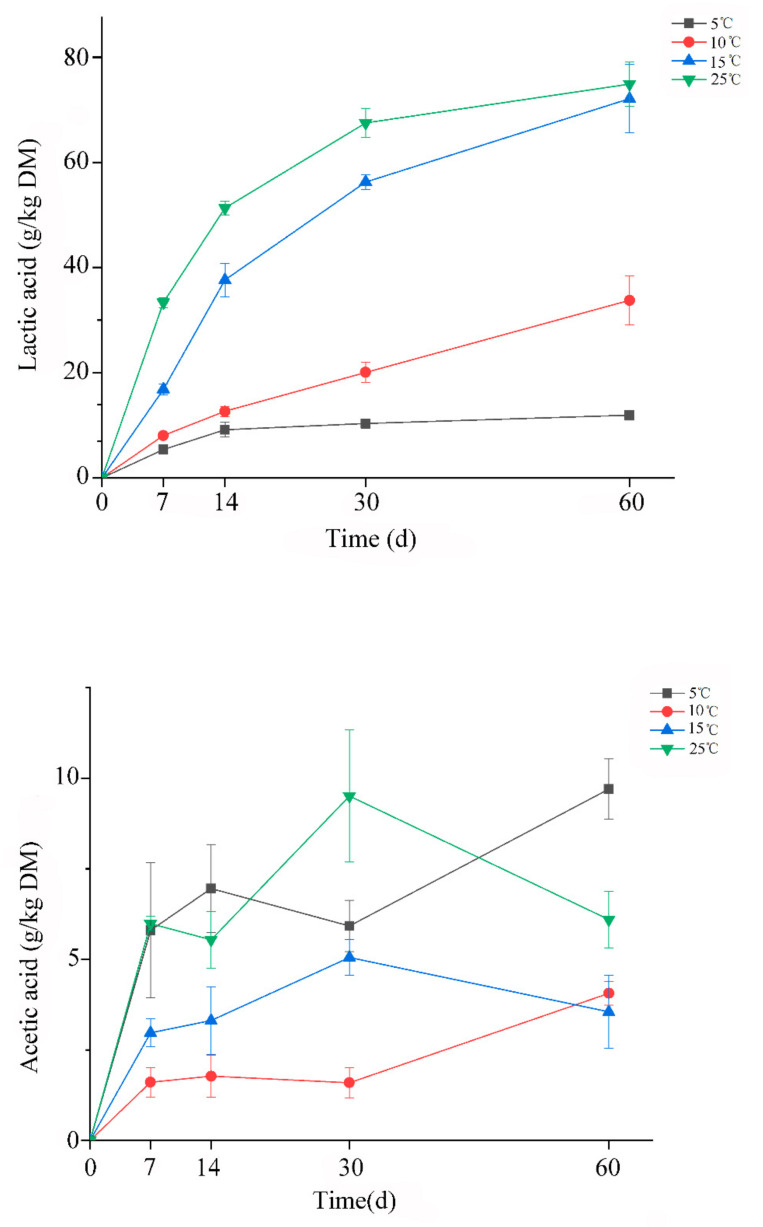
The change of organic acid for oat silages at different temperatures.

**Figure 3 microorganisms-09-00274-f003:**
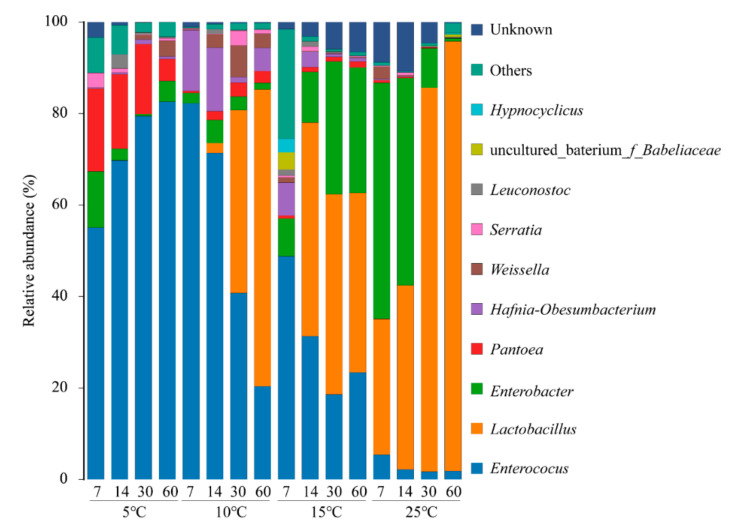
The microbial community of oat silages at the genus level revealed by single molecule, real-time sequencing technology (SMRT).

**Figure 4 microorganisms-09-00274-f004:**
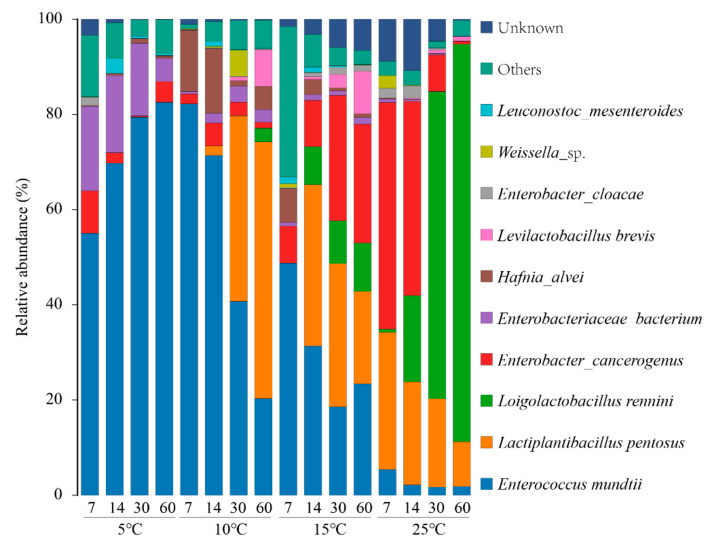
The microbial community of oat silages at the species level revealed by SMRT Cell.

**Figure 5 microorganisms-09-00274-f005:**
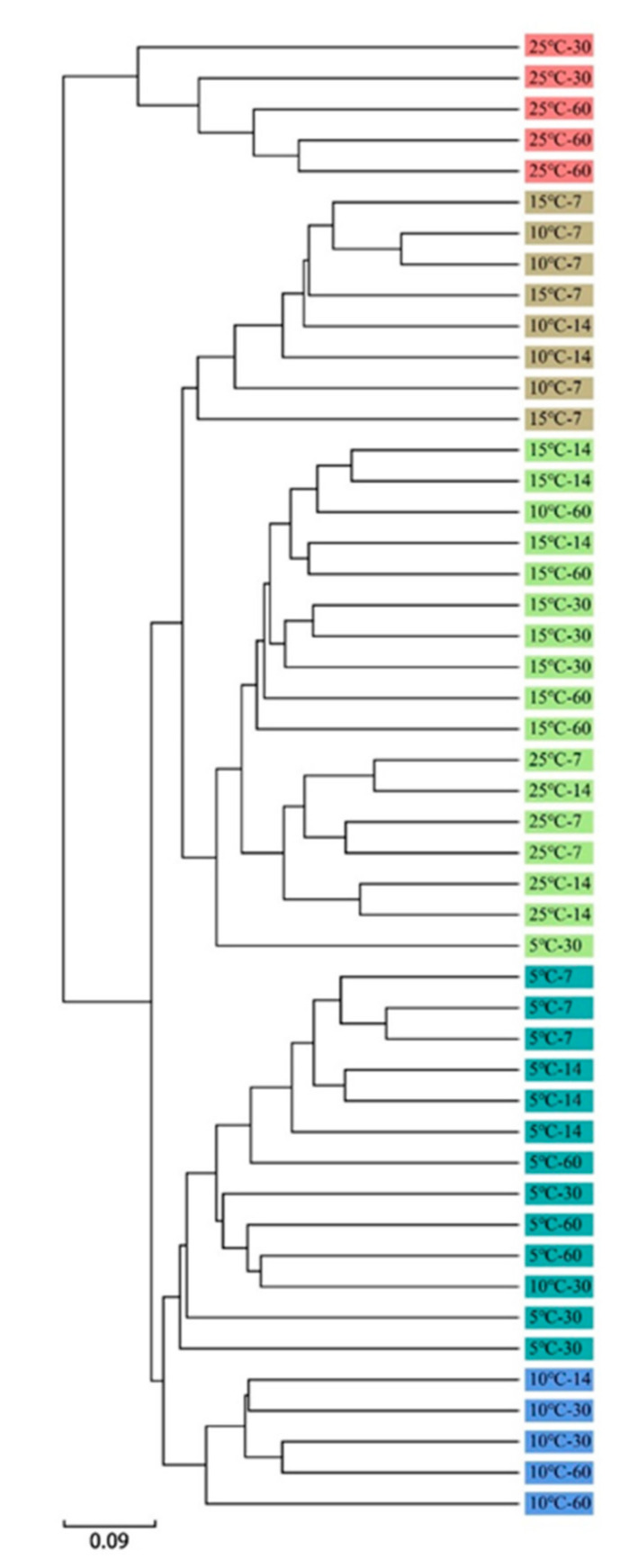
The cluster analysis of microbial community of oat silages based on OTUs.

**Figure 6 microorganisms-09-00274-f006:**
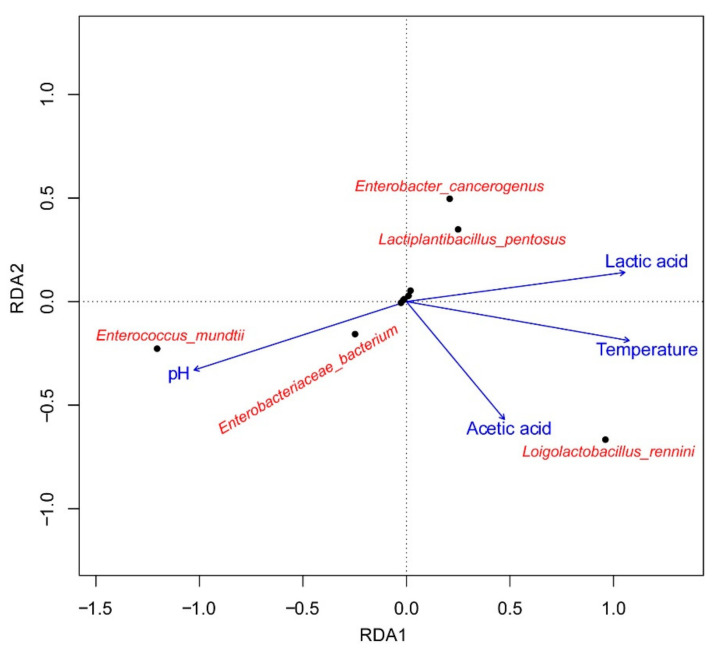
The redundancy analysis (RDA) analysis of high-throughput sequencing data related to the OTU level and major fermentation products. Five bacterial species showing most relationships with fermentation products were used in this analysis.

**Table 1 microorganisms-09-00274-t001:** Chemical composition and microbial population of fresh oat.

Items	Fresh Oat
Dry matter (%)	23.70 ± 1.31
Crude protein (%DM)	13.25 ± 1.19
Acid detergent fiber (%DM)	32.80 ± 1.07
Neutral detergent fiber (%DM)	49.64 ± 2.04
Organic matter (%DM)	95.23 ± 3.20
Ether extract (%DM)	5.12 ± 0.13
Lactic acid bacteria (Log_10_ cfu/g FM^−1^)	3.61 ± 0.12
Yeasts (Log_10_ cfu/g FM^−1^)	4.37 ± 0.23
Aerobic bacteria (Log_10_ cfu/g FM^−1^)	5.12 ± 0.19
Mold (Log_10_ cfu/g FM^−1^)	ND
Coliform	4.1 ± 0.03

DM, dry matter; FM, fresh matter; ND, not detected.

**Table 2 microorganisms-09-00274-t002:** Chemical composition and microbial population of oat silage after 60 days of ensiling.

Temperature	DM(%)	Chemical Composition (%DM)					Microbial Population (Log_10_ cfu/g FM^−1^)				
		CP	ADF	NDF	OM	EE	LAB	Yeast	Aerobic Bacteria	Mold	Coliform
5 °C	27.87	11.82 ^b^	29.29 ^b^	45.40 ^b^	94.34	5.34	5.51 ^a^	ND	6.02	ND	3.16 ^b^
10 °C	26.96	11.70 ^b^	31.73 ^a^	47.05 ^a^	94.22	5.33	5.62 ^a^	ND	7.21	ND	3.21 ^b^
15 °C	25.85	12.68 ^a^	32.00 ^a^	47.49 ^a^	93.56	5.04	5.84 ^a^	ND	6.55	ND	4.45 ^a^
25 °C	25.71	12.41 ^a^	31.27 ^a^	47.19 ^a^	94.45	5.23	4.25 ^b^	ND	6.43	ND	3.08 ^b^
SEM	1.58	0.26	0.86	1.56	0.88	0.23	0.53	−	0.56	−	0.88
Significant analysis	NS	*	*	NS	NS	NS	*	−	NS	−	*

^a,b^ Means in the same column followed by different letters differ (* *p* < 0.05). ND, not detected; NS, not significant; FM, fresh matter; DM, dry matter; CP, crude protein; NDF, neutral detergent fiber; ADF, acid detergent fiber; OM, organic matter; EE, ether extract; LAB, lactic acid bacteria; SEM, Standard Error of Mean.

**Table 3 microorganisms-09-00274-t003:** Alpha diversity of oat silages at different temperatures during ensiling process.

Temperature	Day	OTU	Shannon	Chao	Coverage
5 °C					
	7	51^a^	1.51 ^b^	61 ^c^	0.99
	14	48 ^a^	1.08 ^bc^	67 ^bc^	0.99
	30	31^c^	0.55 ^c^	57 ^c^	0.99
	60	32 ^c^	0.82 ^bc^	78 ^b^	0.99
10 °C					
	7	40 ^b^	0.63 ^c^	57 ^c^	0.99
	14	41 ^b^	1.04 ^bc^	65 ^c^	0.99
	30	49 ^a^	1.34 ^b^	75 ^a^	0.99
	60	51 ^a^	1.45 ^b^	88 ^a^	0.99
15 °C					
	7	60 ^a^	2.10 ^a^	71 ^b^	0.99
	14	45 ^b^	1.79 ^b^	55 ^c^	0.99
	30	51 ^a^	1.81 ^b^	70 ^b^	0.99
	60	50 ^a^	1.89 ^b^	87 ^a^	0.99
25 °C					
	7	40 ^b^	1.43 ^b^	58 ^c^	0.99
	14	36 ^bc^	1.43 ^b^	44 ^d^	0.99
	30	30 ^c^	0.94 ^c^	61 ^c^	0.99
	60	32 ^c^	0.71 ^c^	77 ^b^	0.99
SEM		5.67	0.09	3.17	0.00
Significant analysis:					
Temperature (T)		NS	*p* < 0.05	NS	NS
Day (D)		*p* < 0.05	*p* < 0.05	*p* < 0.05	NS
T×D		*p* < 0.05	*p* < 0.05	NS	NS

^a–d^ Means in the same column followed by different letters differ (*p* < 0.05).; NS, not significant.

## Data Availability

Data is contained within this article.

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
