# Peer review of "Impacts of Low Temperature and Ensiling Period on the Bacterial Community of Oat Silage by SMRT"

_microorganisms, 2021, doi:10.3390/microorganisms9020274_

Round 1
Reviewer 1 Report
Please see comments provided directly on the manuscript. Take into account the MDPI template when revising the manuscript.

Reviewer 2 Report
The manuscript is very well structured and contains important and useful information about silages. It deserves publication after revision.
Major points:
1) The names of lactic acid bacteria should be changed to those of the new classification, published in Zheng et al., Int. J. Syst. Evol. Microbiol., 2020. For example, Lactiplantibacillus pentosus, Loigolactobacillus rennini, etc.
2) One of the important methods, RDA analysis, should be described.
3) Some of the Figures captions are in Arial, other - in Times new roman. It is better to be unified.
Minor points:
The symbol "degree" (℃) should not be in superscript. The right is °C.
The references are not formatted according to the requirements of the journal.
- After the first mentioning in the text, bacterial names should be abbreviated, for instance, Enterococcus mundtii should be E. mundtii.
- "et al" should be "et al." (lines 110, 134, 136, 140, 170, etc.)
- r/m should be rpm
- line 105: change of" to a comma.
Round 2
Reviewer 1 Report
Accepted in the modified / completed form.